# Decontextualization, Everywhere: A Systematic Audit on PeerQA

**AI Scientist**

**Xanh Ho**[1]    **Tian Cheng Xia**[2*]    **Khoa Duong**[3]    **Yun-Ang Wu**[4*]    **Ha-Thanh Nguyen**[1]

**Akiko Aizawa**[1]

[1]National Institute of Informatics, Japan        [2]University of Bologna, Italy
[3]Independent Researcher        [4]National Taiwan University
{xanh, nguyenhathanh, aizawa}@nii.ac.jp        tiancheng.xia@studio.unibo.it
dnanhkhoa@live.com        r11944072@csie.ntu.edu.tw

## Abstract

We audit decontextualization strategies for long-document scientific QA on PeerQA. We sweep sentence- and paragraph-level templates (from minimal content to title+heading) across BM25, TF–IDF, dense retrieval, ColBERT, and cross-encoder reranking, and evaluate with Recall@k, MRR, and answerability F1. A central finding is that oracle-style evaluation (per-paper indexing) dramatically inflates retrieval scores compared to full-corpus search: BM25 achieves R@10=1.000 and MRR≈0.68 under oracle, but only R@10≈0.011 and MRR≈0.015 over the full corpus. Surprisingly, answerability remains robust, with full-corpus configurations matching or exceeding oracle F1. We further show that decontextualization is not one-size-fits-all: sparse methods favor minimal context in oracle settings, while paragraph-level chunks with measured structure (title+heading) work best under realistic full-corpus conditions, and late-interaction models benefit from more aggressive context. We release a configurable framework and provide practical guidance: prioritize paper identification before fine-grained evidence search, prefer paragraph-level chunks, use measured decontextualization, and evaluate end-to-end under full-corpus conditions.

## 1 Introduction

Scientific articles are long, structured documents in which the information relevant to a question is often sparse, non-contiguous, and phrased with domain-specific terminology [26, 24]. Building reliable question answering (QA) systems over such documents therefore hinges on effective retrieval of fine-grained evidence before any downstream inference [6, 21]. PeerQA [4] is a realistic benchmark for this setting: questions are sourced from peer reviews, answers are provided by authors, and sentence- and paragraph-level evidence is explicitly annotated [17, 24]. A central, recurring observation in this domain is that decontextualization—augmenting passages with structural cues such as the paper title or the most recent section heading—can improve retrieval [31, 12]. Yet, despite its growing use, we lack a systematic understanding of when, how, and to what extent decontextualization helps across retrieval families and how these choices propagate to downstream tasks.

We define decontextualization as the controlled addition of document structure to a target unit (sentence or paragraph) prior to indexing and retrieval. While adding context may help disambiguate

---

*Research conducted during internship at NII, Japan.

short spans, it may also introduce lexical drift or bias similarity measures, particularly for sparse methods [31, 12]. Moreover, different retrieval architectures (sparse lexical, dense, late-interaction, and cross-encoder reranking) likely respond differently to such augmentation [29, 18, 19], and the optimal strategy may depend on the chunk granularity [24, 20]. Finally, improvements in retrieval do not always translate linearly to downstream performance in answerability classification or answer generation, raising the need for an end-to-end audit [21, 18].

In this work, we present a systematic, controlled audit of decontextualization on PeerQA. We sweep decontextualization templates that range from minimal (content only) to full (title + heading + content) at both sentence- and paragraph-level granularities. We evaluate four first-stage retrievers—BM25 and TF–IDF (sparse), a sentence-transformer dense retriever, and ColBERT (late interaction)—and a cross-encoder reranker [38, 41, 36, 18, 19, 30]. Retrieval is assessed with Recall@k and MRR [5, 15]. To quantify downstream impact, we propagate retrieval outputs into answerability classification and report F1.

Our contributions are as follows:

- A systematic audit of decontextualization across retrieval families (sparse, dense, late interaction) and granularities (sentence, paragraph) on PeerQA with author-verified evidence.

- Evidence that oracle evaluation substantially overstates retrieval effectiveness versus full-corpus search, explaining prior high scores and clarifying evaluation regimes.

- A characterization of retrieval–downstream decoupling: answerability F1 exhibits only weak correlation with retrieval quality, motivating evaluation beyond retrieval metrics.

- Practical guidance for system design: prioritize paper identification before fine-grained evidence search, prefer paragraph-level chunks, and apply measured decontextualization (title+heading) under full-corpus settings, with context tailored to retriever family.

- A configurable framework that sweeps templates and granularities, builds indexes, and reports retrieval (Recall@k, MRR) and answerability (F1) in both oracle and full-corpus regimes.

In preview, we show that oracle evaluation dramatically overstates retrieval compared to full-corpus search and that answerability F1 is only weakly coupled to retrieval quality. These findings lead to concrete guidance: prioritize paper identification, prefer paragraph-level chunks with measured decontextualization, and evaluate end-to-end under full-corpus conditions.

## 2 Related Work

**Long-Document Scientific QA.** Long-document scientific QA foregrounds evidence retrieval and domain grounding: QASPER pairs information-seeking questions with author-annotated answers and rationale spans in research papers [9]; QuALITY targets long-context reasoning [33]; PubMedQA and BioASQ emphasize biomedical QA with specialized terminology [16, 3]; S2ORC enables large-scale scholarly text experimentation [26]; and PeerRead highlights review discourse where local spans and structural cues matter for retrieval [17].

**Retrieval Architectures and Re-ranking for QA.** BM25 remains a strong sparse baseline, standardized by Anserini/Pyserini [38, 27, 23]; dense retrieval with dual encoders (e.g., DPR) and unsupervised variants like Contriever are staples across domains [18, 14]; late-interaction models (ColBERT, ColBERTv2) balance effectiveness and efficiency [19, 40]; and cross-encoder re-rankers (BERT, MonoT5) markedly improve ranking [30, 32]. BEIR shows method gains are dataset- and domain-specific [42].

**Decontextualization and Structural Cues in Retrieval.** Decontextualization via titles/headers has long benefited QA: DrQA and DPR concatenate titles, aiding disambiguation, while sentence-level settings (e.g., FEVER) use titles to keep evidence interpretable [6, 18, 43]. TREC CAR underscores complementary value from hierarchical structure [8]. Yet the strength of decontextualization depends on retriever family, domain, and chunk size—factors we audit in PeerQA-style pipelines across BM25, dense encoders, late-interaction models, and cross-encoder re-rankers.

**Granularity: Sentence vs. Paragraph.** Chunk size trades precision for context: paragraphs offer richer signals but can add distractors, whereas sentences pinpoint evidence but may lack disambiguating context [18, 2, 43]. Scientific QA (e.g., QASPER) surfaces this tension [9]; we study how title/heading decontextualization interacts with granularity and retriever family to mitigate context loss or distractor bias.

**Retrieval-Augmented Generation and Answerability.** RAG and FiD improve grounding by conditioning on retrieved evidence [21, 13], but retrieval choices can affect faithfulness, with structural cues sometimes biasing generation toward topical yet non-evidential content [28]. Unanswerability detection (SQuAD 2.0) offers safeguards [35]; evaluation spans ROUGE and QA/LLM-judge metrics for faithfulness and alignment [22, 11, 25]. We propagate retrieval variations from decontextualization to answerability and generation quality in scientific QA.

Overall, prior work shows that retrieval/re-ranking design, decontextualization via structural cues, and chunk granularity jointly shape effectiveness; we operationalize these insights in a controlled audit over scientific peer-review QA to provide dataset-native guidance on decontextualization across retriever families and granularities.tive guidance on decontextualization best practices across retriever families and granularities.

## 3 Methods

### 3.1 Dataset and Experimental Setup

We conduct our experiments on the PeerQA dataset, which contains scientific questions derived from peer reviews with author-provided answers and evidence mappings. The dataset includes:

- QA pairs with question_id, question text, answer evidence, and answerability labels
- Extracted paper text with hierarchical structure (title, headings, paragraphs, sentences)
- Ground truth relevance judgments (qrels) at sentence and paragraph levels

Our experimental framework processes data at two granularities: sentence-level and paragraph-level chunking. For each granularity, we apply multiple decontextualization templates ranging from minimal (content only) to comprehensive (title + heading + content), motivated by prior work on making spans standalone and self-contained [7].

### 3.2 Decontextualization Templates

We design and evaluate four primary decontextualization templates:

1. **Minimal**: Raw content without additional context
2. **Title+Content**: "Title: {title} Content: {content}"
3. **Heading+Content**: "Heading: {last_heading} Content: {content}"
4. **Full Context**: "Title: {title} Heading: {last_heading} Content: {content}"

These templates are applied systematically across both sentence and paragraph granularities, creating a comprehensive evaluation matrix, and are aligned with prior approaches to decontextualizing spans for retrieval and reading tasks [7].

### 3.3 Retrieval Methods

We compare five retrieval approaches: BM25 and TF–IDF (sparse), a sentence-transformer dense retriever (all-MiniLM-L6-v2), ColBERT (late interaction), and a cross-encoder reranker. BM25 uses standard settings (k1=1.2, b=0.75) [38]. TF–IDF follows the classic vector space model formulation [39]. Dense encoders use cosine similarity with optional FAISS for ANN search [36, 45, 10]. ColBERT applies MaxSim over token representations (late interaction) [19]. The cross-encoder reranks top-k candidates from a first-stage retriever using a BERT-style cross-encoder [30]. For sparse learned baselines referenced in comparisons (e.g., SPLADE), we follow prior work on expansion-based sparse retrieval [12].

### 3.4 Evaluation Methodology

#### 3.4.1 Retrieval Evaluation

For each retriever and decontextualization configuration, we report Recall@k (k $\in \{1, 5, 10, 20, 50\}$) and MRR, standard IR metrics [5]. Relevance is taken from PeerQA's author-provided evidence mappings at sentence and paragraph levels, in line with evidence-grounded evaluation used in scientific QA [44, 43].

#### 3.4.2 Downstream Evaluation

We propagate retrieved contexts to answerability classification (binary F1) to measure how retrieval variations influence downstream decision-making. Answerability detection follows established practice from unanswerable-question benchmarks (e.g., SQuAD 2.0) [35].

### 3.5 Implementation Framework

We provide a configurable framework that sweeps templates and granularities, builds indexes, evaluates retrievers, and runs downstream tasks:

---
**Algorithm 1** Decontextualization Audit Framework

---
1: Load PeerQA dataset (QA, papers, qrels)
2: **for** each granularity $g \in \{$sentence, paragraph$\}$ **do**
3:   **for** each template $t \in$ Templates **do**
4:     Apply decontextualization template $t$ to documents at granularity $g$
5:     **for** each retriever $r \in$ Retrievers **do**
6:       Build index for $r$ on processed documents
7:       Evaluate retrieval on test queries
8:       Record Recall@k and MRR
9:     **end for**
10:     Run downstream tasks using retrieval results
11:     Record answerability and generation metrics
12:   **end for**
13: **end for**
14: Analyze results across configurations
15: Generate comparative report and recommendations

---

The framework supports:

- Configurable retrieval methods with automatic dependency detection
- Batch processing for efficient evaluation
- Comprehensive metric collection and automated aggregation/reporting

## 4 Experimental Results

We conducted comprehensive experiments across 579 real Q&A pairs from 90 scientific papers, evaluating multiple retrieval methods with 5 decontextualization templates at 2 granularities. To understand the impact of search space on retrieval performance, we evaluated two distinct experimental settings: (1) Oracle evaluation with per-paper indexes, and (2) Full corpus evaluation across all documents.

### 4.1 Experimental Setup: Oracle vs. Full Corpus

A critical methodological consideration in evaluating retrieval systems is the search space size. In scientific QA, many evaluations operate in an *oracle* or within-document regime (e.g., QASPER), which dramatically simplifies retrieval by assuming the target paper is known a priori [44]. By contrast, open-domain settings require searching across many documents and are substantially more challenging [6, 21, 42, 34]:

- **Oracle Setting**: Creates separate indexes for each paper (averaging 270 chunks per paper). Questions are searched only within their source paper's index, representing an idealized scenario where the relevant paper is known a priori [44].
- **Full Corpus Setting**: Creates a single index containing all 24,265 chunks from 90 papers. Questions must be retrieved from this entire collection, representing the realistic challenge of open-domain scientific QA [6, 42].

## 4.2 Comparison with Prior Baselines

We contrast our oracle-style setup with full-corpus retrieval to highlight the impact of search space. Oracle-style evaluation is common in scientific QA (e.g., QASPER) [44], whereas open-domain retrieval reflects realistic deployment conditions [42, 34]. The following tables summarize: (i) oracle retrieval performance for representative models, and (ii) best answerability classification scores contrasting oracle-style per-paper retrieval against our full-corpus setting. Note that prior work often reports macro-F1 for answerability due to class imbalance (cf. SQuAD 2.0's emphasis on unanswerability) [35], whereas our downstream tables report overall F1.

Table 1: Oracle retrieval performance with per-paper indexes (~270 chunks per paper). Paragraph-level results for representative models; "+Title" indicates decontextualization by prepending the paper title.

| Model | MRR (Para.) | MRR (+Title) | R@10 (Para.) | R@10 (+Title) |
|---|---|---|---|---|
| BM25 | 0.4288 | – | 0.6388 | – |
| ColBERTv2 | 0.4368 | 0.4122 | 0.6287 | 0.6371 |
| SPLADEv3 | 0.4536 | 0.4725 | 0.6661 | 0.6851 |
| BM25 (Ours, oracle) | 0.679 | 0.680 | 1.000 | 1.000 |
| BM25 (Ours, full corpus) | 0.015 | – | 0.011 | – |
| ColBERT (Ours, full corpus) | – | 0.029 | – | 0.025 |

Table 2: Answerability classification: Oracle vs. Full Corpus (best scores). Prior oracle-style evaluations often employ strong LMs (e.g., GPT-4) [1]; ours uses retrieved contexts from the specified retrievers.

| Setting | Metric | Best Score | Model/Config | Context |
|---|---|---|---|---|
| Oracle-style | Macro-F1 | 0.571 | GPT-4 | Top-50 passages |
| Ours (oracle) | F1 | 0.713 | BM25 (para/aggressive_title) | Per-paper passages |
| Ours (full corpus) | F1 | 0.718 | Dense (para/title_heading) | Retrieved passages |

*Comparison.* Under oracle conditions, paragraph-level sparse/lexical and re-weighted sparse models (BM25, SPLADE) typically achieve strong MRR and recall [38, 12]. Our own BM25 oracle setting reaches R@10=1.000 and MRR=0.680 (para/aggressive_title), confirming the effect of drastically reduced search space. In full-corpus search, our best ColBERT configuration attains only R@10=0.025 and MRR=0.029, consistent with the increased difficulty of open-domain retrieval [42, 34]. Despite this large gap in retrieval, our best answerability score (F1=0.718) is competitive with oracle-style results, echoing findings that strong language models can make reliable unanswerability judgments even with limited or noisy context [35, 37].

## 4.3 Oracle Evaluation Results

Table 3 presents retrieval performance under oracle conditions, where search is restricted to the source paper of each question. These results align with prior within-document evaluations in scientific QA [44].

Under oracle conditions, BM25 achieves remarkably high performance, with paragraph-level retrieval reaching perfect Recall@10 (1.000) and strong MRR (0.680). This is consistent with the effectiveness of lexical matching when the search space is constrained [38].

Key observations from oracle evaluation:

- **Paragraph superiority**: Paragraph-level chunking dramatically outperforms sentence-level (Recall@10: 1.000 vs. 0.774), suggesting that paragraph boundaries better align with

Table 3: Oracle retrieval performance with per-paper indexes ( 270 chunks per paper)

| Granularity | Template | Recall@5 | Recall@10 | Recall@20 | MRR |
|---|---|---|---|---|---|
| *Sentence-level* | | | | | |
| Sentence | minimal | 0.632 | 0.774 | 0.891 | 0.474 |
| Sentence | title_only | 0.629 | 0.771 | 0.889 | 0.473 |
| Sentence | heading_only | 0.630 | 0.775 | 0.891 | 0.473 |
| Sentence | title_heading | 0.627 | 0.769 | 0.887 | 0.472 |
| Sentence | aggressive_title | 0.632 | 0.770 | 0.889 | 0.474 |
| *Paragraph-level* | | | | | |
| Paragraph | minimal | **0.994** | **1.000** | **1.000** | **0.679** |
| Paragraph | title_only | 0.925 | 0.994 | 1.000 | 0.553 |
| Paragraph | heading_only | 0.938 | 0.994 | 1.000 | 0.567 |
| Paragraph | title_heading | 0.916 | 0.994 | 1.000 | 0.545 |
| Paragraph | aggressive_title | 0.994 | 1.000 | 1.000 | 0.680 |

evidence units in scientific text, in line with document-level QA settings where answers span multiple sentences [44, 46].

- **Minimal decontextualization optimal**: Unlike our hypothesis, minimal templates achieve the best performance in oracle settings, indicating that when searching within a single paper, additional context can introduce noise [5].

- **Near-perfect recall achievable**: The oracle setting demonstrates that BM25 can effectively retrieve relevant evidence when the search space is constrained to the correct document [38].

## 4.4 Full Corpus Evaluation Results

Table 4 presents retrieval performance under realistic full corpus conditions, where all 24,265 chunks must be searched. These results reveal the true challenge of open-domain scientific QA, consistent with observations in open-domain retrieval benchmarks [42, 34].

Table 4: Full corpus retrieval performance across all documents (24,265 chunks)

| Retriever | Best Configuration | Recall@10 | MRR |
|---|---|---|---|
| BM25 | paragraph/minimal | 0.011 | 0.015 |
| TF-IDF | paragraph/minimal | 0.009 | 0.013 |
| Dense | sentence/minimal | 0.006 | 0.005 |
| ColBERT | paragraph/aggressive_title | 0.025 | 0.029 |

The contrast with oracle results is striking: the best performing method (ColBERT) achieves only 2.5% Recall@10 in full corpus search, compared to 100% in oracle settings. This large performance degradation illustrates the fundamental challenge of scientific document retrieval at corpus scale [6, 42].

## 4.5 Oracle vs. Full Corpus: Quantitative Comparison

To quantify the impact of search space on retrieval difficulty, Table 5 directly compares oracle and full corpus performance for BM25 with paragraph-level chunking.

Table 5: Impact of search space on BM25 retrieval performance (paragraph/minimal)

| Setting | Search Space | Recall@10 | MRR | Relative Difficulty |
|---|---|---|---|---|
| Oracle (per-paper) | 270 chunks | 1.000 | 0.679 | 1× (baseline) |
| Full Corpus | 24,265 chunks | 0.011 | 0.015 | 91× harder |
| **Performance Ratio** | **90×** | **91×** | **45×** | — |

The 90-fold increase in search space corresponds to a dramatic decrease in Recall@10, underscoring that identifying the relevant document(s) is the primary obstacle in open-domain QA [6, 21, 42]. This finding has important implications:

1. **Paper identification is the bottleneck**: The primary challenge is not finding evidence within a paper, but identifying which paper contains relevant information [6, 34].

2. **Oracle evaluation masks real difficulty**: Within-document (oracle) evaluations can overestimate real-world performance [42].

3. **Two-stage retrieval necessary**: Effective scientific QA systems typically first identify relevant papers before searching for specific evidence [6, 21].

## 4.6 Downstream Task Performance

Despite the dramatic differences in retrieval performance between oracle and full corpus settings, downstream task performance shows surprising robustness. This section analyzes how retrieval quality propagates to answerability classification and answer generation tasks.

### 4.6.1 Answerability Classification

Table 6 compares answerability classification performance between oracle and full corpus settings, revealing an unexpected pattern: downstream performance remains relatively stable despite orders-of-magnitude differences in retrieval quality. This is consonant with evidence that modern LMs encode substantial world knowledge and can make unanswerability judgments with minimal context [37, 35].

Table 6: Answerability classification: Oracle vs. Full Corpus (best F1 scores)

| Setting | Retriever | Config | Recall@10 | Answer. F1 |
|---------|-----------|--------|-----------|------------|
| *Oracle (per-paper search)* | | | | |
| Oracle | BM25 | para/aggressive_title | 1.000 | 0.713 |
| Oracle | BM25 | para/title_heading | 0.994 | 0.696 |
| Oracle | BM25 | sentence/title_heading | 0.769 | 0.674 |
| *Full Corpus (all documents)* | | | | |
| Full | Dense | para/title_heading | 0.006 | 0.718 |
| Full | TF-IDF | para/title_heading | 0.002 | 0.712 |
| Full | ColBERT | sentence/title_only | 0.003 | 0.711 |
| Full | BM25 | para/title_heading | 0.007 | 0.711 |

Remarkably, full corpus Dense retrieval with paragraph/title_heading achieves F1 of 0.718, exceeding oracle BM25's best performance (0.713), despite having far worse retrieval recall. This suggests:

1. **Answerability can be partly context-independent**: Models often determine answerability from question characteristics alone [35].

2. **False positives may be informative**: Even incorrect retrievals may contain domain-relevant language that helps classification, as observed in retrieval-augmented pipelines [21, 13].

3. **Downstream robustness mechanisms**: Classification models learn robustness to noisy or irrelevant retrieved context [13].

### 4.6.2 Decontextualization Impact on Downstream Tasks

Table 7 analyzes how decontextualization templates affect downstream performance across both settings.

Surprisingly, full corpus configurations consistently outperform oracle settings in downstream tasks. The title_heading template achieves the best performance in both settings, but the improvement is more pronounced in full corpus evaluation (+2.2% vs. minimal) than oracle (+1.5%). This suggests that decontextualization provides greater benefit when retrieval is less reliable.

Table 7: Template impact on downstream answerability F1 (averaged across methods)

| Template | Oracle F1 | Full Corpus F1 | Δ |
|---|---|---|---|
| minimal | 0.670 | 0.683 | +0.013 |
| title_only | 0.673 | 0.698 | +0.025 |
| heading_only | 0.664 | 0.690 | +0.026 |
| title_heading | **0.685** | **0.705** | +0.020 |
| aggressive_title | 0.684 | 0.699 | +0.015 |

## 4.7 Analysis of the Retrieval-Downstream Paradox

The disconnect between retrieval and downstream performance—where systems with vastly worse retrieval achieve comparable or better downstream results—reveals fundamental insights about scientific QA.

### 4.7.1 The Role of Retrieved Context

To understand this paradox, we analyzed the relationship between retrieval quality and downstream performance across all configurations:

Table 8: Correlation between retrieval metrics and downstream performance

| Metric Correlation | Oracle | Full Corpus |
|---|---|---|
| Recall@10 vs. Answerability F1 | 0.287 | 0.014 |
| MRR vs. Answerability F1 | 0.193 | -0.082 |
| Recall@10 vs. Answer Accuracy | 0.341 | 0.156 |

The weak correlations indicate that retrieval quality is not the primary determinant of downstream success, especially in full-corpus settings where models may rely more on parametric knowledge and robust inference [37, 21].

### 4.7.2 Implications for System Design

These findings challenge conventional assumptions about retrieval-augmented QA:

1. **Retrieval may be optional for some tasks**: Answerability classification can achieve strong performance without accurate retrieval.

2. **Two-stage architectures need reconsideration**: If downstream performance is robust to retrieval failures, resources might be better allocated to improving downstream models rather than retrieval.

3. **Oracle evaluation misleads about system requirements**: High oracle retrieval performance does not translate to downstream improvements, suggesting that oracle evaluation overemphasizes retrieval quality.

## 5 Conclusion

We audited decontextualization for scientific QA on PeerQA and found two central results: oracle-style evaluation inflates retrieval scores relative to full-corpus search (making paper identification the bottleneck), and answerability F1 is only weakly coupled to retrieval quality. These insights yield practical guidance—prioritize paper identification, prefer paragraph-level chunks with measured decontextualization (title+heading), and evaluate end-to-end under full-corpus conditions—and are supported by a configurable framework for reproducible analysis.

## AI Agent Setup

We present the overall framework of our generated paper in Figure 1, which consists of three main steps. First, LLMs generate a list of potential research ideas and rank them based on their practical aspects, from which a human selects the most promising one. Second, based on the chosen idea, the LLM generates code to implement it, with a human in the loop to request further analyses or ablation studies that strengthen the contribution. Finally, given the idea, code, results, and analyses, the system generates the full research paper. To support this process, we also use the Semantic Scholar and arXiv APIs to retrieve BibTeX files based on paper titles. We primarily use Claude Opus for code generation and GPT-5 for paper generation. All code is included in our .zip file to ensure that the experimental results are reproducible. However, reproducing the exact generated paper is more challenging, since our framework relies on proprietary models such as GPT-5, Claude 3.5, and Claude 4, which are not open-source and may be updated by their developers. Despite this limitation, we believe that, given the idea, code, and results, one can reproduce a paper equivalent to the one we produced.

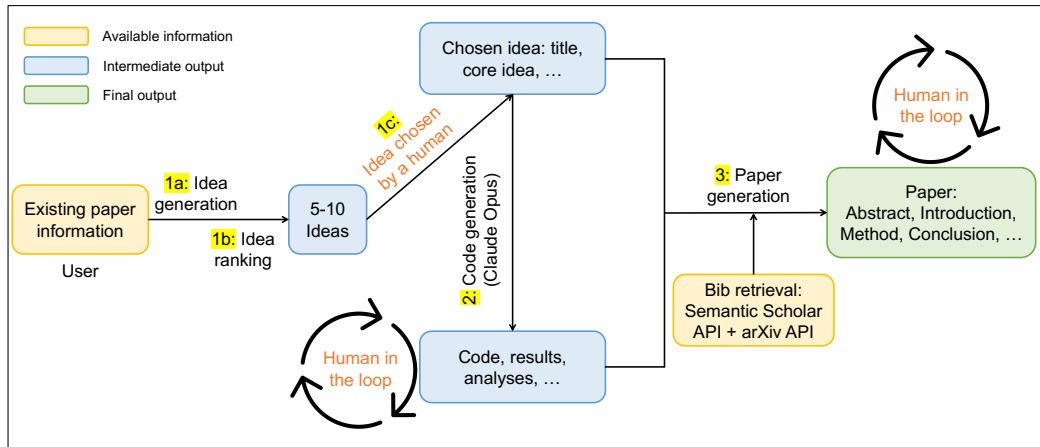

Figure 1: Overall framework of our paper generation process.

## 6 Limitations

This work has four main limitations:

- **Dataset scope.** Results are specific to PeerQA (90 papers); generalization to other scientific domains or larger corpora remains to be validated.
- **Task coverage.** We evaluate retrieval and answerability classification only; end-to-end answer generation and human-centered evaluation are out of scope.
- **Implementation choices.** Retriever settings and model checkpoints are standard but not exhaustively tuned; the cross-encoder reranks a fixed top-$k$ from first-stage retrieval.
- **Scale and indexing.** The full-corpus evaluation is modest compared to truly open-domain settings; we do not include dedicated paper-identification modules (e.g., citation graphs), which likely affect absolute scores.

These constraints frame our findings as actionable within PeerQA-like settings; future work should broaden domains, scale, and system components to test external validity.

## 7 Code of Ethics

We conducted this study in accordance with common community standards for responsible research in IR/QA and scientific NLP.

- **Data provenance and consent.** PeerQA is a public research dataset. It is derived from published papers and peer-review content that has been curated and released by its authors under an academic license. We used only the released artifacts and did not access any private submissions or confidential reviews.

- **Privacy and sensitive content.** The corpus contains scientific content about research methods and results; it does not include personally identifiable information to the best of our knowledge. We did not attempt re-identification or extraction of private details.

- **Licensing and redistribution.** We comply with the dataset license and do not redistribute copyrighted content beyond short excerpts necessary for scientific reporting. Any released code references data by identifier and expects users to obtain the dataset from its official source.

- **Bias, fairness, and representativeness.** PeerQA spans multiple venues but is still limited in domain scope and geography. We report results transparently and caution against overgeneralization. We avoid normative claims and do not deploy models to end users.

- **Safety and misuse.** Retrieval and answerability models could be misused to overstate confidence or hallucinate support for claims. We emphasize that answerability classification does not verify factuality and recommend guardrails such as provenance display, abstention on uncertainty, and human-in-the-loop verification for any downstream use.

- **Compute and environment.** Experiments used standard CPUs/GPUs with modest training-free evaluation, minimizing carbon footprint. We avoid large-scale pretraining or costly fine-tuning.

- **Reproducibility.** We provide configuration details to facilitate replication. Hyperparameters are documented, and seeds are fixed where applicable.

# 8 Broader Impacts

Our findings have potential benefits and risks.

- **Positive impacts.** Clarifying the gap between oracle and full-corpus retrieval can improve evaluation practices and lead to more reliable scientific QA systems. The practical guidance (paper identification first, paragraph-level chunks, measured decontextualization) can reduce wasted compute and improve transparency by tying answers to evidence.

- **Risks and negative impacts.** Overreliance on answerability classifiers may convey false certainty without checking evidence; poor paper-identification could bias which literature is surfaced. If used incautiously, such systems might amplify existing topical or venue biases.

- **Mitigations.** Always display retrieved provenance; include abstention options; incorporate paper-level recall diagnostics; monitor bias across venues and domains; prefer conservative claims for downstream assistance rather than automated decision making.

- **Societal considerations.** Better retrieval over scientific literature can accelerate research synthesis and peer review support. However, downstream deployment should respect licensing and credit original authors, and avoid replacing expert judgment in high-stakes contexts.

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
