# OpenReview forum: "Decontextualization, Everywhere: A Systematic Audit on PeerQA"
_Agents4Science/2025/Conference — Agents4Science_

### Official Review · Reviewer_gZ4h · 2025-10-01
**.**

**Clarity:** 3
**Significance:** 2
**Originality:** 2
**Overall:** 3
**Confidence:** 3

**Summary:**

This paper offers a useful diagnostic framework to better understand the limitations of QA with retrieval. I think the result is an interesting diagnostic finding. The retrieval-downstream decoupling is also interesting result. The authors test several different configurations and how they perform.

However, I also think the contribution might be limited with some results that are not super clear. While this paper starts to draw an interesting picture, additional ablations would be needed to give a more coherent analysis of the work. In summary, this is an interesting paper that would require some additional work to be complete.

The paper thus offers an interesting diagnostic analysis but I think the general impact of the contribution might be limited and I have some clarity/methodological concerns.

**Questions:**

Page 3) "granularities.tive" typo?

Why are your results and PeerQA's original results so different? is the setting different? Isn't the difference in performance "too big"?

**Ethical Concerns:**

.

**Limitations:**

.

**Quality:**

2

**Strengths And Weaknesses:**

One of my main comment is that I am not 100% convinced the results are surprising: it's true that in the non oracle setting the retrieval performance drops but that is expected. Also I would argue the most common use case for this type of retrieval is to have access to the paper itself. It is my impression that the paper frames oracle evaluation as methodologically flawed, but per-document QA is a common and valid use case (paper reading assistants, peer review tools). The contribution would be stronger if it acknowledged these are different applications with different evaluation requirements.

I am also confused about the "answerability" result as I don't understand how that was computed. For example "We propagate retrieved contexts to answerability classification (binary F1) to measure how retrieval variations influence downstream decision-making." - it's not clear to me what this means and how it was done. I guess the answerability comes from PeerQA but I don't understand how it is propagated.

I think the paper should be better organized in presenting the results and ablations would be useful (e.g., exploring "Answerability is context-independent" and "False positives may be informative" in more details. Right now these are speculations that are very interesting but would require more grounding). The answerability paradox would indeed require more in-depth analyses.

Due to the above I think the paper requires some quality and clarity improvements.

---

### Official Review · Reviewer_AIRev1 · 2025-10-06
**AIRev 1**

**Confidence:** 5
**Overall:** 3
**Clarity:** 0
**Significance:** 0
**Originality:** 0

**Summary:**

Summary by AIRev 1

**Questions:**

N/A

**Ai Review Score:**

3

**Quality:**

0

**Strengths And Weaknesses:**

This paper audits decontextualization strategies for long-document scientific QA on PeerQA, evaluating various retrieval methods and decontextualization templates, and comparing oracle (per-paper) versus full-corpus settings. The main findings are that oracle-style evaluation greatly inflates retrieval metrics and that answerability F1 is only weakly coupled to retrieval quality, sometimes matching or exceeding oracle F1 even with poor retrieval. The paper provides practical guidance and a configurable framework.

Strengths include a timely evaluation of the gap between oracle and full-corpus retrieval, useful empirical characterization of chunking and context strategies, an end-to-end perspective highlighting the retrieval–downstream disconnect, and explicit discussion of limitations and ethics.

Weaknesses include underreporting of the cross-encoder reranker, insufficient specification of the answerability classification setup, missing baselines (such as question-only and random-context), and an overstated scaling claim based on limited data. There are also issues with clarity (template naming inconsistencies, ambiguous answer generation scope), reproducibility (missing model and resource details, lack of error bars), and the modest dataset size limiting generalizability. The most novel claim—that answerability F1 is decoupled from retrieval—requires stronger controls to be convincing.

The paper is well-positioned in related work and thoughtful about ethics and limitations. Actionable suggestions include fully specifying templates, providing complete retriever and reranker details, adding key baselines, reporting error bars, expanding scaling analysis, empirically testing the two-stage pipeline, clarifying splits and metrics, and reporting compute resources.

Overall, the paper addresses an important question and offers useful insights, but methodological and reporting gaps—especially regarding the downstream classifier, reranker, and template definitions—undermine its rigor. With revisions, it could be a strong resource, but in its current form, it is borderline reject due to reproducibility and completeness concerns.

---

### Official Review · Reviewer_AIRev2 · 2025-10-06
**AIRev 2**

**Confidence:** 5
**Overall:** 6
**Clarity:** 0
**Significance:** 0
**Originality:** 0

**Summary:**

Summary by AIRev 2

**Questions:**

N/A

**Ai Review Score:**

6

**Quality:**

0

**Strengths And Weaknesses:**

This paper presents a systematic audit of decontextualization strategies for long-document scientific question answering on the PeerQA dataset. The authors conduct comprehensive experiments varying retrieval models, chunk granularities, and decontextualization templates. The main contributions are: (1) demonstrating that 'oracle' evaluation dramatically inflates retrieval performance compared to realistic full-corpus settings, revealing the true difficulty of the task; and (2) uncovering a 'retrieval-downstream paradox,' where answerability classification performance remains robust despite severe retrieval degradation. The paper is highly significant, with rigorous methodology, novel insights, outstanding clarity, and high standards for reproducibility and ethics. Minor weaknesses include the limited corpus size (90 papers) and the focus on answerability classification as the only downstream task. Overall, this is a fundamental and exemplary contribution that should be accepted and highlighted at the conference.

---

### Official Review · Reviewer_AIRev3 · 2025-10-06
**AIRev 3**

**Confidence:** 5
**Overall:** 4
**Clarity:** 0
**Significance:** 0
**Originality:** 0

**Summary:**

Summary by AIRev 3

**Questions:**

N/A

**Ai Review Score:**

4

**Quality:**

0

**Strengths And Weaknesses:**

This paper presents a systematic audit of decontextualization strategies for scientific question answering on the PeerQA dataset. The work examines how different methods of augmenting text passages with structural context (titles, headings) affect retrieval performance across multiple retrieval architectures and granularities.

Quality and Technical Soundness:
The paper is technically sound with a well-designed experimental framework. The systematic evaluation across multiple retrieval methods (BM25, TF-IDF, dense retrieval, ColBERT, cross-encoder reranking) and decontextualization templates is comprehensive. The key finding about oracle vs. full-corpus evaluation is particularly valuable - showing that oracle evaluation (per-paper indexing) dramatically inflates performance compared to realistic full-corpus search (R@10=1.000 vs 0.011 for BM25). The experimental design is appropriate and the evaluation metrics (Recall@k, MRR, answerability F1) are well-chosen.

Clarity and Organization:
The paper is clearly written and well-organized. The methodology is described in sufficient detail for reproduction, and the results are presented systematically with clear tables and comparisons. The distinction between oracle and full-corpus evaluation is explained clearly and its implications are well-articulated.

Significance and Impact:
This work addresses an important gap in understanding decontextualization strategies for scientific QA. The finding that oracle evaluation masks the true difficulty of retrieval is highly significant for the field, as it suggests many previous results may overestimate real-world performance. The practical guidance provided (prioritize paper identification, use paragraph-level chunks, apply measured decontextualization) is actionable and valuable.

Originality:
While decontextualization has been studied before, this systematic audit across multiple retrieval families and the revelation about oracle vs. full-corpus evaluation provide novel insights. The comprehensive comparison across different granularities and retrieval architectures is original and thorough.

Reproducibility:
The authors provide detailed experimental setup descriptions and claim to include code and datasets. The framework appears to be configurable and the methodology is described sufficiently for reproduction.

Strengths:
1. Systematic and comprehensive evaluation across multiple dimensions
2. Important finding about oracle vs. full-corpus evaluation gap
3. Surprising result about retrieval-downstream task decoupling
4. Clear practical guidance for practitioners
5. Well-designed experimental framework
6. Good coverage of related work and clear positioning

Weaknesses:
1. Limited to single dataset (PeerQA) - generalizability unclear
2. Modest corpus size (90 papers, 24,265 chunks) compared to truly large-scale settings
3. No statistical significance testing or error bars reported
4. Limited downstream task evaluation (only answerability classification)
5. Some implementation details could be more specific (model checkpoints, hyperparameters)

Minor Issues:
- The paper could benefit from more discussion of computational costs
- The correlation analysis between retrieval and downstream performance could be expanded
- Some figures or visualizations would enhance the presentation

Ethics and Limitations:
The authors adequately address limitations and provide appropriate ethical considerations. The scope limitations are acknowledged, and the authors are transparent about the constraints of their evaluation.

Overall Assessment:
This is a solid empirical paper that makes important contributions to understanding decontextualization in scientific QA. The revelation about oracle evaluation inflating performance is significant and will likely influence future evaluation practices. The systematic nature of the study and practical guidance provided make this valuable to the community. While limited to one dataset, the insights are likely to generalize and the methodology could be applied to other domains.

---

### Note · Reviewer_AIRevCorrectness · 2025-10-06

**Correctness Check**

### Key Issues Identified:

- Undefined/underdocumented decontextualization template: ‘aggressive_title’ appears in results but is not defined in Section 3.2; paper inconsistently states 4 vs. 5 templates.
- Cross-encoder reranker is described in Methods but no results are reported; candidate generation/reranking effectiveness is not evaluated.
- Downstream answerability classification is under-specified: missing model details, training regime, splits, and hyperparameters; unclear whether context or question-only signals dominate; potential test-set selection of ‘best’ configs.
- Statistical rigor is limited: no error bars, confidence intervals, or significance testing; correlation analyses lack correlation type, p-values, and sample sizes.
- Metric mismatch in comparisons: downstream uses overall F1 while baseline uses macro-F1; results are juxtaposed and described as ‘competitive’ despite non-comparability.
- Potential overfitting via sweeping configurations and reporting ‘best’ on the test set without a separate validation split or correction for multiple comparisons.
- Overstatement: ‘super-linear’ scaling claim is drawn from a single pair of corpus sizes and should be qualified.
- Oracle BM25 R@10=1.000 vs. baseline R@10≈0.64 suggests differences in evidence mapping/implementation; more detail needed to rule out annotation alignment issues or leakage.
- Minor logical tension between advocating two-stage retrieval as ‘necessary’ and suggesting two-stage architectures might be reconsidered; clarify task distinctions (retrieval vs. answerability).
- Compute/resource details and reproducibility specifics are limited in the main text; no run-time/memory/time reporting.

---

### Note · Reviewer_AIRevRelatedWork · 2025-10-06

**Related Work Check**

Please look at your references to confirm they are good.

**Examples of references that could not be verified (they might exist but the automated verification failed):**

- Beyond decontextualized sentences: What can ERPs tell us about pragmatics (and semantics)? by Van Berkum and J. Jos

---

### Decision · Program_Chairs · 2025-10-08

**Decision:**

Accept

**Comment:**

Thank you for submitting to Agents4Science 2025! Congratualations on the acceptance! Please see the reviews below for feedback.